# Molecular Biology and Therapeutic Perspectives for K-Ras Mutant Non-Small Cell Lung Cancers

**DOI:** 10.3390/cancers14174103

**Published:** 2022-08-24

**Authors:** Elona Cekani, Samantha Epistolio, Giulia Dazio, Marco Cefalì, Luciano Wannesson, Milo Frattini, Patrizia Froesch

**Affiliations:** 1Oncology Institute of Southern Switzerland (IOSI), Ospedale Regionale di Bellinzona e Valli, Ente Ospedaliero Cantonale (EOC), 6500 Bellinzona, Switzerland; 2Laboratory of Molecular Pathology, Institute of Pathology, Ente Ospedaliero Cantonale (EOC), 6601 Locarno, Switzerland; 3Oncology Institute of Southern Switzerland (IOSI), Ospedale Regionale di Lugano—Civico e Italiano, Ente Ospedaliero Cantonale (EOC), 6900 Lugano, Switzerland; 4Oncology Institute of Southern Switzerland (IOSI), Ospedale Regionale di Locarno, Ente Ospedaliero Cantonale (EOC), 6600 Locarno, Switzerland

**Keywords:** *KRAS*, lung adenocarcinoma, *KRAS* biology, prognosis, clinical relevance, immune checkpoint inhibitors

## Abstract

**Simple Summary:**

Mutations in the Kirsten rat sarcoma viral oncogene homolog (*KRAS*) gene are the most common alterations in non-small cell lung cancer (NSCLC); they are generally linked to smoking history and are typical of the lung adenocarcinoma (AC) subtype. The clinical relevance of *KRAS* mutations in NSCLC was generally low or null until a few years ago. What is now emerging is that *KRAS*-mutant lung AC patients are generally associated with poorer survival. However, the approval of Sotorasib, a KRAS inhibitor, improved the survival of a previously untargetable *KRAS*-mutant lung AC group. Furthermore, new approaches targeting *KRAS* are under development. Starting from the description of the biology of *KRAS*-mutant NSCLC, the present review focuses on the clinical aspects of *KRAS* mutations in NSCLC. Finally, the interaction between *KRAS* mutations and immune checkpoint inhibitors will also be discussed.

**Abstract:**

In non-small cell lung cancer (NSCLC) the most common alterations are identified in the Kirsten rat sarcoma viral oncogene homolog (*KRAS*) gene, accounting for approximately 30% of cases in Caucasian patients. The majority of mutations are located in exon 2, with the c.34G > T (p.G12C) change being the most prevalent. The clinical relevance of *KRAS* mutations in NSCLC was not recognized until a few years ago. What is now emerging is a dual key role played by *KRAS* mutations in the management of NSCLC patients. First, recent data report that *KRAS*-mutant lung AC patients generally have poorer overall survival (OS). Second, a *KRAS* inhibitor specifically targeting the c.34G > T (p.G12C) variant, Sotorasib, has been approved by the U.S. Food and Drug Administration (FDA) and by the European Medicines Agency. Another KRAS inhibitor targeting c.34G > T (p.G12C), Adagrasib, is currently being reviewed by the FDA for accelerated approval. From the description of the biology of *KRAS*-mutant NSCLC, the present review will focus on the clinical aspects of *KRAS* mutations in NSCLC, in particular on the emerging efficacy data of Sotorasib and other *KRAS* inhibitors, including mechanisms of resistance. Finally, the interaction between *KRAS* mutations and immune checkpoint inhibitors will be discussed.

## 1. Introduction

The Rat Sarcoma virus (*RAS*) gene family is characterized by three genes (*KRAS*, *NRAS* and *HRAS*) encoding for closely related GTPase proteins responsible for the signal transduction within the mitogen-activated protein kinase (MAPK) and phosphatidylinositol 3 kinase (PI3K) pathways. This group of genes is associated with the control of cell growth, survival and differentiation [1,2], and it is characterized by the most frequently observed genomic alterations in human cancers, being detected in approximately 19% of tumors [3,4].

*KRAS* mutations are the most important and frequent alterations in this family, accounting for 85% of *RAS* mutations observed in the oncologic population [5]. More specifically, *KRAS* exon 2, exon 3 and exon 4 mutations are present in 13.98%, 1.17% and 0.62% of all cancer types [2,6], respectively, especially in pancreatic ductal adenocarcinoma (PDAC) (88% of patients), in colorectal cancer (CRC) (45–50% of cases) and in lung adenocarcinoma (AC) (30–35% of patients) (Table 1) [5,6,7,8].

Despite the fact that *KRAS* is altered in a consistent number of cases, the clinical interest in assessing *KRAS* mutations in lung was generally low or null until a few years ago; it has gained great relevance in the last few years [9,10], with the development of several targeted approaches and the recent introduction of sotorasib into clinical practice. This review will therefore summarize the current state-of-the-art of the clinical relevance of *KRAS* mutations as well as the most recent advances in KRAS-targeted anti-cancer strategies, focusing in particular in lung AC. Furthermore, the main mechanisms of acquired resistance to KRAS G12C inhibitors will be outlined and the attempts to overcome this resistance will be reported.

## 2. *KRAS* Mutations in Lung Cancer

The protein encoded by *KRAS* is a GTPase regulated by tyrosine kinase receptors that in turn activate the RAF/MEK/MAPK and PI3K/AKT signaling pathways, leading to a strong promotion of cell growth and replication that determines cell survival. The presence of mutations in *KRAS* blocks this protein in the active GTP-bound conformation, thereby activating constitutively downstream signaling pathways, finally resulting in uncontrolled cell growth [11]. Specifically, this continuous activation confers to cancer cells the ability to grow in lower glucose concentrations than those required for the growth of normal cells or cancer cells that do not have *KRAS* mutations [12,13]. As for other cancer types, *KRAS* mutations in lung cancers typically occur in hot-spot codons of exons 2, 3 and 4.

### 2.1. KRAS Mutations in NSCLC

In general, lung cancer has a complex and heterogeneous background. Indeed, it can be histologically subdivided into non-small cell lung cancer (NSCLC) and small cell lung cancer (SCLC), accounting for 85% and 15% of all lung neoplasia, respectively [14]. NSCLCs are further subdivided into three histologic types: AC, squamous-cell carcinoma (SCC) and large-cell carcinoma (LCC), which represent 40%, 25–30% and 5–10% of all lung cancers, respectively [8]. The occurrence of KRAS mutations differs based on the specific lung cancer type/subtype, being higher in NSCLC than in SCLC, and within NSCLC, KRAS mutations are more frequent in lung AC than SCC (Table 2). Interestingly, a recent study established two different groups of KRAS-mutant NSCLC, KRAS-dependent and KRAS-independent according to their requirement for KRAS mutations to maintain oncogenicity. However, to date, these data are preliminary and need to be confirmed in larger cohorts [11].

#### 2.1.1. KRAS Mutations in Lung Adenocarcinoma

In the AC subtype, almost 30–35% of cases are driven by a *KRAS* mutation [5,6,7,8], mainly in exon 2, with the c.34G > T (p.G12C) change being the most prevalent (Table 3) [15].

#### 2.1.2. KRAS Mutations in Squamous-Cell Carcinoma

The percentage of SCC-harboring *KRAS* mutations is lower than in lung AC cases, corresponding to only 3–4% of patients [7,16,17]; again, the c.34G > T (p.G12C) change as the most prevalent one, identified in up to 41% of *KRAS*-mutant SCC patients (Table 2) [18].

## 3. *KRAS* Mutations in Population-Based Cohorts of Lung Adenocarcinomas

The frequency of *KRAS* mutation in patients affected by lung AC differs among ethnicities and countries of origin, being higher in Caucasian (where they are detected in up to 30% of patients) than in Asians or in Africans (occurring in 16% and 9% of cases, respectively) [19,20,21,22,23,24,25].

### 3.1. KRAS Mutations in Caucasians Affected by Lung AC

In Europe, the most frequent *KRAS* mutations in lung AC are found in exon 2, with a general mutational rate close to 30% [19,20,21]. Differences can also be observed among European countries (28% in Germany, 29–31% in France, 27% in Italy, 19–27% in Greece and 43.3% in Spain) [19,20,21,26,27,28,29] and even among regions of the same country: for example, the island of Sardinia in Italy has the lowest frequency among all Italian regions [30]. This difference could be due to the physical separation of the island from the country, reflecting different lifestyle or environmental behaviors.

In Europe, 86% of mutations in exon 2 are located in codon 12, with the c.34G > T (p.G12C) mutation being the most common one (observed in 34% of cases) [19] (Table 4).

### 3.2. KRAS Mutations in Asians Affected by Lung AC

Several studies show that Asian populations have significant lower *KRAS* mutation incidence than that in the Caucasian and Afro-American populations [37,38]. As for Europe, even in Asia there is a wide range in the mutation frequencies among countries, with the highest values in Vietnam (21%, close to European populations) and the lowest in China (8.4–13%), Japan (10%) and Indonesia (7%) [33,39,40,41,42]. As usual, the most affected region is *KRAS* exon 2, but the most common *KRAS* exon 2 mutational subtype is the c.35G > A (p.G12D) change, with a rate of 25.5%, followed by c.34G > T (p.G12C) (24.5%) [22,32,43] (Table 4).

The frequency and type of mutations in exons 3 and 4 in Asian populations mirror those observed in European countries [43] (Table 4).

### 3.3. KRAS Mutations in Africans Affected by Lung AC

Central and West Africa are the regions with the lowest incidence rates of lung cancer for both sexes (0.8%) worldwide [44], while in North Africa, the *KRAS* mutations is observed in 6–9% of patients [31,36,45]. Specific studies concerning molecular characterization in African populations are extremely rare, and the data are sporadic. In Morocco, the c.34G > T (p.G12C) change is the most prevalent one (detected in 73% of *KRAS*-mutant cases) [31], while in Tunisia, c.38G > A (p.G13D) followed by c.35G > A (p.G12D) are predominant (42.8% and 28.5% of cases, respectively) [36] (Table 4). However, the size cohort is often too small to derive any conclusion.

### 3.4. KRAS Mutations in the Heterogeneous Lung AC American Population

In the USA, *KRAS* mutations are observed in a significantly high proportion of cases, up to 47.5% of patients, especially in exon 2 (90.6%), with the c.34G > T (p.G12C) change being the most prevalent one (41.2%) [34]. Focusing on the different ethnicities, the prevalence of *KRAS* mutations in African-American patients was less than that observed in the Caucasian group (17% versus 26%) [46]. Interestingly, American Hispanic patients have better survival than non-Hispanic/non-African cases, despite the fact that this population is diagnosed for cancer at a later stage due to social factors. This situation, defined also as the “Hispanic paradox”, has suggested investigating Hispanic individuals’ biological, pathological and molecular factors [35]. First, researchers observed a lower frequency of *KRAS* mutations compared with the Caucasian population (15.7% and 20–35% respectively) [35], and, second, researchers described that Hispanics have more *KRAS* changes in exon 2 codon 13 and in exon 3 codon 59 than non-Hispanics [47], possibly explaining the inferior cancer aggressiveness.

In Latin America, the frequency of NSCLC (in particular lung AC) with *KRAS* mutations corresponds to 14% [48], with little differences, ranging from 12.9% in Colombia to 15.9% in Mexico and 16.8% in Peru [48].

Two relevant observations come from analyses of Puerto Rican and Brazilian cohorts [49,50]. In Puerto Rico, lung ACs present with *KRAS* mutations in 18.7% of the cases, a rate that is significantly higher than in the aforementioned general Hispanic populations [48,50]; in Brazil, 14.6% of lung ACs harbour *KRAS* mutations, with a higher prevalence in non-Asian patients [49]: in both populations, c.34G > T (p.G12C) is the most diffused type of alteration.

## 4. Risk Factors and *KRAS* Mutations in Lung Adenocarcinomas

Recently, the number of lung cancer cases has increased with different trends in men versus women [51]. In women, lung cancer incidence rates began to increase significantly in 1973 and reached a plateau in the late 1990s, over a decade later than men [52,53]. Women’s lung cancer mortality stabilized for the first time in 2003, two decades later than men, and has yet to decline [53,54]. In 2021, the estimated new cases for males and females were nearly comparable (119,100 new patients versus 116,660, respectively) [51].

“Women’s emancipation” and the strategy of marketing cigarettes directly to women and young girls both contributed to increasing the percentages of female smokers and consequently to rising rates of lung cancer in women [55]. Hormonal influences, reduced DNA damage reparability and elevated amount of DNA adduct are also important factors in the high percentage of women who suffer from lung cancer [56].

However, among all the factors correlated with the development of lung tumors, tobacco smoking is sufficient to increase the risk of lung cancer compared with never smokers and represents the predominant cause, accounting for 80% of all lung cancer cases [57]. In the meta-analysis conducted by Chapman and his group [58], never-smokers present a distinct and separate mutational pattern of lung cancer from that observed in smokers. Smokers are characterized by more complex *KRAS* mutations, have a higher mutational burden and often present co-occurring mutations in *TP53* or *STK11* compared with cancers occurring in never smokers [59]. *STK11* is a tumor suppressor encoding for the serine/threonine kinase STK11, also known as liver kinase B1 (LKB1), and it regulates cell polarity and energy metabolism [60]. In addition, it has been proposed that *KRAS* mutations might be more common in smoking females than in smoking males [61], a finding not confirmed by subsequent studies [62]. Smoking-associated *KRAS* mutations are mainly transversion mutations (G > T and G > C) and are more frequent in females than in males [61,63,64]: In fact, the two most common *KRAS* mutations in NSCLC are the c.34G > T (p.G12C) and the c.35G > T (p.G12V) changes [65,66], which are more common in females and in current or past smokers [62]. In contrast, *KRAS* transition mutations (G > A) are more common in non-smoker patients, suggesting different mechanisms of carcinogenesis [63,64].

Genetic factors are also associated with a higher risk of developing lung cancer. A meta-analysis involving the results of 32 studies demonstrated that most non-smokers with a familial history of lung cancer have a 2-fold increase probability of developing this type of malignancy [67]. In more detail, individuals with a first-degree relative affected by lung cancer have a 1.5 higher probability of developing this disease. This risk is even greater when a sibling is affected by lung cancer [68]. Nevertheless, no specific correlation between familiarity and *KRAS* mutations has been reported because *KRAS* mutations in lung AC are somatic mutations and not linked to an inherited predisposition [69].

Brenner [70] found that nonsmokers who suffer from previous respiratory diseases such as chronic obstructive pulmonary disease (COPD), asthma, pneumonia and tuberculosis are at higher risk of developing lung cancer. However, COPD is not associated with the presence of *KRAS* mutations in lung cancer [61].

Additional risk factors associated with lung cancer development are viral infections, improper lifestyle habits such as obesity, dietary factors, alcohol, consumption of marijuana and environmental air pollution. To the best of our knowledge, none of these factors has been associated with the occurrence of *KRAS* mutations [71,72,73,74,75,76,77,78,79,80].

In conclusion, only tobacco smoking is strictly related to the occurrence of *KRAS* mutations in lung cancer patients.

## 5. *KRAS* Biology in Lung Cancer

As was aforementioned, mutations in the *KRAS* gene primarily cause the constitutive activation of the downstream signaling pathways, leading to uncontrolled cell proliferation and abnormal cell survival [81,82]. Mutant allele specific imbalance (MASI) has been observed in combination with *KRAS* mutation in 58% of lung cancers and has been demonstrated to be mainly caused by uniparental disomy, which results from the complete loss of the wild-type (wt) allele without copy number gain [83]. Additionally, Soh et al. found a significant association between *KRAS* mutations and MASI with increased GTP-bound active RAS protein [83].

An in vitro analysis conducted on cell lines carrying different *KRAS* codon subtypes found that the c.35G > A (p.G12D) mutation was associated with PI3K and MEK phosphorylation, while *KRAS* c.34G > T (p.G12C) and *KRAS* c.35G > T (p.G12V) were correlated with Ral pathway activation and subsequently with AKT phosphorylation reduction [84].

As described before, cigarette smoking has been associated with *KRAS* mutations, but alterations in this gene were also related to abnormal metabolic parameters such as hypertension, dyslipidemia and diabetes mellitus [57,85]. This might indicate a relationship of the *KRAS* mutational status with metabolic parameters. However, this hypothesis was rejected by Yilmaz and colleagues [86].

## 6. *KRAS* and Co-Occurring Mutations in Lung AC

Much evidence suggests that *KRAS*-mutant cancers, including lung ACs, are a heterogeneous disease composed of different cellular clones and subclones carrying specific molecular patterns. These biological diversities can be ascribed to the presence of co-occurring genetic events, to the co-presence of different *KRAS* gene mutation subtypes and to the mutant *KRAS* allelic content [87]. Although the role of *KRAS* co-occurring mutations remains to be elucidated, it has been described that they may affect treatment responses [88].

### 6.1. Co-Occurring Genetic Events

#### 6.1.1. KRAS and Other Molecular Markers Alterations

*KRAS* mutations are mutually exclusive with alterations in other molecular markers such as *EGFR*, *ALK* and *ROS1*, considered genetic drivers of lung cancer development. However, rare cases (<1%) harboring mutations in both *KRAS* and these markers have been reported [88]. In lung ACs, the rate of co-occurring mutations with *KRAS* alterations is 3.2%, and of the *KRAS*-mutant cases, 33% [88,89]. These frequencies differ based on the specific *KRAS* mutation: c.34G > T (p.G12C) presents reportable co-mutations in 6.1% of lung ACs, c.35G > A (p.G12D) in 2.5% and c.35G > T (p.G12V) in 1.6% (including co-existing alterations with *EGFR*, *BRAF* and *NRAS*) [87,88]. The most frequent *KRAS* co-mutations were found in *TP53* (42%), *STK11* (29%) and Kelch-like ECH-associated protein 1 (*KEAP1*)/Nuclear factor erythroid 2-related factor 2 *(NFE2L2)* genes (27%) [90]. This is relevant because Keap1, interacting with Nrf2, a transcription factor encoded by the *NFE2L2* gene, regulates the antioxidant response that is involved in the amelioration of oxidative stress [91].

Recently, the presence of mutations in different genes as well as concomitant mutations in the same gene have been investigated as a factor influencing targeted therapy responses or patients’ prognosis [88]. Although they rarely occur in lung cancer, *KRAS* mutations combined with *EGFR* mutations or *ALK* rearrangement predict poorer response to tyrosine kinase inhibitors (TKIs) [88].

Concerning co-occurring genetic events, an integrative analysis of genomic, transcriptomic and proteomic data in lung ACs identified three subsets of *KRAS*-mutant cases. The first one was characterized by co-occurring alterations in *KRAS/STK11/LKB1* (KL group), the second by mutations in *KRAS/TP53* (KP group) and the third by mutations in *KRAS* plus CDKN2A/B inactivation coupled with low expression of the NKX2-1 (TTF1) transcription factor (KC group) [92]. These groups differ in biology, response to therapies and prognosis (please see chapter 15 for specific details).

#### 6.1.2. KRAS and PD-L1 Expression

Preclinical evidence indicates that PD-L1 level is elevated through the activation of downstream KRAS-signaling pathways [93]. Consequently, many studies investigated the association between *KRAS* mutations and PD-L1 expression. Liu and coauthors described that *KRAS*-mutant tumors were more likely to be PD-L1 positive and showed more T-cell infiltration than *KRAS* wt tumors (*p* = 0.0002 and *p* = 0.003) [93,94,95]. In addition, a higher proportion of combined PD-L1 expression and CD8+ tumor infiltrating lymphocyte (TILs) presence was observed in the *KRAS*-mutant group compared with the wt (*p* = 0.008). The percentage of *KRAS*-mutant cases also harboring PD-L1 expression is equal to 49.5% considering PD-L1 ≥ 1% or to 21.2% considering PD-L1 ≥ 50% [94]. These high frequencies suggest how *KRAS* mutations lead to an inflammatory phenotype [93]. In deepening the investigation of the different *KRAS* mutations subtypes, a higher proportion of PD-L1-positive tumors (≥50%) has been found in patients with c.35G > A (p.G12D), accounting for 41.7%; c.35G > T (p.G12V) accounted for 33.3%; and c.37G > T (p.G13C) accounted for 40%, whereas on the lower end, the rates were c.34G > T (p.G12C) accounting for 9.8% and c.35G > C (p.G12A) accounting for 14.3% [94]. The concomitance between *KRAS* mutations and PD-L1 expression is also related to a longer survival after immunotherapies and as a consequence can be defined as a prognostic and predictive factor of benefit from anti-PD1/PD-L1 immunotherapies [93,95].

The relevance of *KRAS* compared with PD-L1 expression has been further evaluated in smokers comparing *KRAS*-mutant versus wt, with mutant showing higher PD-L1 expression and T-cell infiltration (*p* = 0.042 and *p* = 0.008) [93].

### 6.2. Co-Presence of Different KRAS Gene Mutation

In rare cases (<1%), patients with lung AC can present more than one co-occurring mutation in *KRAS*, usually occurring in two different *KRAS* exons (2 and 3), with the most frequent combination being c.34G > T (p.G12C) or c.35G > A (p.G12D) with c.182A > T (p.Q61L) or c.182A > G (p.Q61R) [88,89]. The double mutations in *KRAS* represent in general 11–26% of the co-occurring genetic mutations, with c.34G > T (p.G12C) accounting for the most diffused mutation associated with other alterations (mainly c.38G > A, p.G13D; c.34G > A, p.G12S; c.35G > T, p.G12V; c.182A > T, p.Q61L; c.182A > G, p.Q61R) in the same gene (nearly 20%) [89].

## 7. *KRAS* Influence on Prognosis

Multiple studies have shown conflicting results regarding the prognostic value of *KRAS* mutations in lung cancer [21,96,97]. Many researchers reported that *KRAS* mutations correlate with shorter survival and shorter time to relapse [98,99,100]. Aredo et al. [101] reported that the *KRAS* c.35G > A (p.G12D) mutation was significantly associated with poor OS, while Nadal and his colleagues [34] found a worse prognosis in individuals harboring the *KRAS* c.34G > T (p.G12C) mutation with resectable disease. In addition, Park [102] demonstrated a low response to the treatment with pemetrexed in patients with the *KRAS* c.34G > T (p.G12C) mutation. Furthermore, the *KRAS* c.35G > T (p.G12V) mutation was correlated with a shortened OS and time to recurrence [20], while Izar et al. [103] showed a superior disease-free survival in patients with resected stage I disease, characterized by *KRAS* c.34G > T (p.G12C) and c.35G > T (p.G12V) mutations. Finally, other studies did not find any correlation between *KRAS* mutations and survival [104,105].

The prognosis associated with *KRAS* mutations depends on co-alterations. A poorer prognosis is found in patients with co-occurring mutations of *KRAS* and *STK11* or *KEAP1* [106,107] or *KRAS* c.34G > T (p.G12C) and PD-L1 expression [105,108]. Regarding the correlation between *KRAS* mutations and PD-L1 expression, a recent contribution of our group reported the potential role played by the *KRAS* c.34G > T (p.G12C) variant in the prolonged survival found in NSCLC patients showing a PD-L1 overexpression [109].

## 8. Predictive Value of *KRAS* Mutations

Many attempts have been made to outline the predictive value of *KRAS* mutations in NSCLC patients treated with standard chemotherapy [110,111,112,113], targeted therapies [114,115] and immune check point inhibition [116].

### 8.1. Predictive Value of KRAS Mutation for Response to Chemotherapy

In spite of recent advances in the treatment of NSCLC, a significant percentage of patients with advanced-stage disease are still treated with platinum-based chemotherapy regimens. *KRAS* mutations as a predictive biomarker in NSCLC were investigated in the neo-adjuvant and perioperative setting [117], in the adjuvant setting [111], and in metastatic disease, with both exclusive conventional chemotherapy and the association of erlotinib or placebo [10,110]. In general, in the above-mentioned experiences, *KRAS* mutations were not predictors of response rate, progression-free survival (PFS) or OS in patients with lung cancer.

The type of chemotherapy regimen may also have to be considered. In a subgroup analysis from the JBR10 trial, the combination of vinorelbine and cisplatin in the adjuvant setting appeared to confer a statistically significant survival benefit in the *KRAS* wt subgroup, although *RAS* status did not appear to have a statistically significant effect on outcomes in the interaction analysis [118]. On the contrary, Sun et al. reported significantly poorer PFS and OS in patients with *KRAS* mutations when treated with pemetrexed or gemcitabine but not in those receiving taxanes [112].

The exact subtype of *KRAS* mutation should be considered when investigating its potential role as a predictive biomarker for chemotherapy in NSCLC patients. A potential detrimental effect of *KRAS* codon 13 mutations was suggested by Garassino et al. [66], and similar results have been reported by other authors who observed that PFS and OS were significantly shorter in patients harboring *KRAS* codon 13 mutations and treated with carboplatin-based adjuvant chemotherapy [97]. The role of the *KRAS* c.35G > T (p.G12V) mutation is controversial. One study showed poorer response to chemotherapy in terms of PFS and disease control rate (DCR) compared with wt patients and non-*KRAS* c.35G > T (p.G12V) metastatic mutated patients [119], while another project demonstrated better response to platinum-based chemotherapy compared with other codon 12 mutations [120].

Preclinical data from Garassino et al. add further complexity, suggesting an interaction between the *KRAS* mutation subtype and the type of chemotherapy used. Specifically, their data show that the c.34G > T (p.G12C) clones are associated with a reduced response to cisplatin and an increased sensitivity to taxol and pemetrexed, whereas the c.35G > A (p.G12D) variant results in resistance to taxol treatment and sensitivity to sorafenib. Furthermore, c.35G > T (p.G12V) mutant clones are associated with a better response to cis-platin compared with the wt clones. On the other hand, these clones appeared to be more resistant to pemetrexed. Compared with other chemotherapy regimens, the sensitivity of tumor cells to gemcitabine does not depend on different subtypes of *KRAS* mutations [66]

### 8.2. Predictive Value of KRAS Mutations for Response to Anti-Angiogenic Therapy

KRAS promotes the initiation of tumor growth and ensures tumor progression by stimulating tumor-associated angiogenesis [121,122] and activating different effector pathways through the upregulation of VEGF (vascular endothelial growth factor) and CXC chemokine interleukin-8 (IL-8) [121,123,124,125,126].

However, very few studies have investigated the influence of *KRAS* mutations on the efficacy of anti-angiogenic therapy [127,128,129].

A phase II trial assessed the efficacy of the addition of bevacizumab to neo-adjuvant chemotherapy in resectable, non-squamous NSCLC and found that no patients with *KRAS* mutation had pathological response to neo-adjuvant bevacizumab combined with chemotherapy, whereas 35% of patients with *KRAS* wt status showed a significant pathological response [114].

As retrospectively shown by Ghimessy et al., patients with *KRAS* mutations, and especially those with *KRAS* c.35G > A (p.G12D) mutant disease, had a significantly shorter OS and PFS compared with those with *KRAS* WT when treated with bevacizumab plus platinum-based chemotherapy [115]. Accordingly, c.35G > A (p.G12D) mutations may define a subset of *KRAS* types that might not be eligible for treatment with anti-VEGF drugs [115].

### 8.3. Predictive Value of KRAS Mutations for Response to EGFR Targeted Therapy

The majority of published data suggest that *KRAS* mutational status is a significant negative predictor of the efficacy of EGFR-directed TKI [10,128,129,130], probably because the *KRAS* mutation can counteract the therapeutic effect of EGFR inhibitors [130]. Indeed, when patients with *KRAS*-mutated disease are treated with EGFR TKIs, they tend toward worse objective response rates (ORR), PFS and OS compared with patients without a *KRAS* mutation [10,130,131].

However, controversies exist, likely in relation to a potential role of the involved *KRAS* mutation subtype since specific subtypes have been shown to induce differential sensitivity to EGFR TKIs [24,132]. Within the *KRAS*-mutant group, Metro and colleagues identified great variability in terms of sensitivity to treatment: KRAS codon 13 mutant patients displayed worse outcomes when compared with KRAS codon 12 mutant patients and KRAS wt patients [133]. Similarly, other researchers demonstrated poorer treatment efficacy in the case of *KRAS* c.34G > T (p.G12C) and c.35G > T (p.G12V) mutations and found promising response rates in c.35G > A (p.G12D) and c.34G > A (p.G12S) *KRAS*-mutant NSCLC treated with EGFR-targeted TKIs [134].

Fiala et al. reported that EGFR-directed TKIs improved PFS in patients with non- c.34G > T (p.G12C) *KRAS*-mutant tumors when compared with the c.34G > T (p.G12C) group [135]. In a meta-analysis, Mao and colleagues showed an ORR of 3% (6/210) in patients with mutant *KRAS* NSCLC and of 26% (287/1125) in *KRAS* wt lung cancers, fulfilling the hypothesis that *KRAS* mutations represent negative predictive biomarkers for tumor response in NSCLC patients treated with EGFR-TKIs [131]. However, due to the substantial mutually exclusive relationship between *KRAS* and *EGFR* mutations, the clinical usefulness of *KRAS* mutation as a selection marker for EGFR-TKIs sensitivity in NSCLC is limited [131].

### 8.4. Predictive Value of KRAS Mutations for Response to IMMUNOTHERAPY in NSCLC

PD-L1 expression has been shown to be in close relationship with *KRAS* status, and *KRAS* mutations were described as possible biomarkers for immune checkpoint inhibitor efficacy [114], in particular the c.34G > T (p.G12C) mutant variant [109,135,136,137,138,139]. Monotherapy of immune checkpoint inhibitors in advanced lines demonstrated a higher objective response rate (26%) in *KRAS*-mutant patients when compared with other oncogene-driven NSCLC patients (BRAF, ROS1, MET, EGFR, HER2, RET, ALK) as reported in immunotarget study [140]. The same clinical trial showed no differences within *KRAS* mutation subtypes in terms of PFS and response rate, although patients with KRAS c34G > T (p.G12C) seemed to have better PFS and response rates compared with non-KRAS c34 > T (p.G12C) patients. [139]. The positive correlation of *KRAS* mutations and ICI efficacy was confirmed in a comprehensive analysis [141] and in several studies including patients with PD-L1 overexpression (≥50%) [94,112,142].

Concerning *KRAS* mutation subtypes, in the retrospective study by Jeanson et al., PD-L1 positivity was significantly higher among patients with *KRAS* c.35G > A (p.G12D), c.35G > T (p.G12V) and c.37G > T (p.G13C) mutations, making these subgroups of patients potential candidates for immunotherapy; meanwhile, PD-L1 negativity was higher among those who had c.35G > C (p.G12A) and c.34G > T (p.G12C) mutations [139].

Overall, the correlation between KRAS mutations and efficacy of ICI seems to be quite robust.

## 9. Current Therapeutic Approaches and Perspectives in *KRAS-Mutant* NSCLC

Because of its high mutation frequency in NSCLC, KRAS is an appealing target. However, the development of targeted therapies for *KRAS*-mutant lung cancers has long been marked by failures [11,143,144]. In contrast to most of the oncogenic drivers, KRAS was for many years considered undruggable because of its extremely high affinity for GTP, the lack of well-defined hydrophobic pockets and the complexity of the downstream pathways [145,146,147]. KRAS inhibition strategies have included reducing the proportion of active RAS-GTP, disrupting protein–protein interactions, decreasing RAS presence at the plasma membrane, inhibiting downstream effector signaling and inducing synthetic lethality.

Initial development focused on preventing RAS from associating with the cell membrane, e.g., through the inhibition of the post-translational farnesylation of its C-terminal domain or through the competitive inhibition of RAS membrane binding; however, these strategies failed to demonstrate clinical efficacy due to the emergence of escape mechanisms such as adaptive KRAS prenylation by geranylgeranyl transferase [148,149,150,151]. Attention therefore turned to strategies for targeting KRAS indirectly by interfering with its upstream, downstream or parallel pathways; however, initial experiences with the inhibition of signaling proteins such as RAF, FAK, MEK, MET, mTOR and AKT met limited success [11,152,153,154].

Unsuccessful attempts were made with synthetic lethality approaches as well, probably due to the presence of either the feedback activation of RAS upstream effectors or tumor heterogeneity [149,155,156,157,158,159].

### 9.1. Direct Inhibition of KRAS

A breakthrough in the direct targeting of KRAS occurred in 2013, when a new allosteric pocket was discovered beneath the effector switch-II region of mutant *KRAS* c.34G > T (p.G12C) [160] and a small molecule (ARS-1620) was identified that could bind covalently to this pocket. Later, a second small molecule with the same activity was discovered (ARS-853) [161].

These studies demonstrated that the direct targeting of KRAS c.34G > T (p.G12C) is dependent on covalent binding to a switch-II pocket region when KRAS c.34G > T (p.G12C) is in its inactive GDP-bound state and to cysteine binding as well: therefore, these drugs do not recognize, and as a consequence are inactive against, other KRAS-mutant or KRAS wild-type proteins. Later preclinical studies allowed for better defining and identifying the mechanisms of action of inhibitors in *KRAS* c.34G > T (p.G12C) mutant cases [160]; [161,162,163]. Subsequently, AMG-510 and MRTX849 were the first inhibitors introduced in a preclinical setting to demonstrate increased *KRAS* c.34G > T (p.G12C) inhibition activity over previous inhibitors, mediated by the enhanced synergic activity of H95 residue in the a-3 helix of *KRAS* c.34G > T (p.G12C) [162,163,164,165].

These results led to the initiation of the first-in-human trial of AMG-510 (sotorasib) in 2018 (ClinicalTrials.gov NCT03600883). Today, several clinical studies are evaluating direct KRAS inhibition using similar drugs (Table 5).

### 9.2. Sotorasib, First KRAS c.34G > T (p.G12C) Inhibitor in Clinical Practice

Sotorasib successfully passed all testing phases, from in vitro KRAS c.34G > T (p.G12C) mutant cell lines to preclinical setting: its activity led to the inhibition of ERK phosphorylation and tumor cell growth, while the PI3K signaling was substantially unaffected [146,166]. Moreover, sotorasib was found to upregulate a pro-inflammatory microenvironment, in this way enhancing anti-tumor T cell activity [164,167] and as a consequence inhibiting a potential active promoter of tumor progression. The repercussion of these positive results was the introduction in a first phase I/II trial of sotorasib as the first drug against KRAS mutation in clinical development. In this first study, sotorasib showed an objective response rate of 32.2% in the subgroup of patients with NSCLC and disease control in 88.1% of patients with a mPFS of 6.3 months [168].

On the post hoc analysis of Codebreak 100 (phase II study), sotorasib demonstrated an ORR of 37.1%, DCR of 80.6% with a median duration of response (DOR) of 11 months, a PFS of 6.8 months and a median OS of 12 months. These outcomes led to an FDA priority review in February 2021 and to an ongoing phase III clinical trial comparing sotorasib with docetaxel for patients with KRAS c34G > T (p.G12C) mutant NSCLC (ClinicalTrials.gov NCT04303780).

The efficacy of sotorasib among patients with brain metastases was also evaluated in post hoc analysis by Ramingam et al. In this subgroup of 40 patients included in a phase I/II Codebreak 100 trial, sotorasib demonstrated clinical efficacy in terms of PFS (5.3 months) and OS (8.3 months). [169]. However, several side effects were also reported: grade 3–4 treatment-related adverse events were reported in nearly 20% of patients, diarrea in 31.7%, nausea in 19%, transaminitis (ALT and AST) in 15% and asthenia in 11.1% of cases. Regarding liver tolerability, it was reported that sotorasib might trigger immune-related hepatitis in patients previously treated with ICIs [170]. Overall, sotorasib maintained or improved global health status, physical functioning and the severity of the main lung-cancer-related symptoms, including cough, dyspnea and chest pain [171]. In late 2021, a randomized study of dose efficacy was prepared, focusing on the comparison of different doses of the drug (240 mg versus 960 mg, with the higher dose being the one that was used in the phase II and III studies) as part of a US FDA post-marketing requirement [172].

Furthermore, it has been demonstrated that cases with *STK11* and *TP53* co-occurring mutations responded better to sotorasib than patients harboring *KEAP1* co-mutation [169,170,171,173,174,175]. Moreover, alterations in cell cycle effectors like in the WNT and MAPK pathways are more common in the late progressors group [176]. As recently reported by Zhao and colleagues, patients with a lower plasma *KRAS* c34G > T (pG12C) allele frequency and a lower tumor burden at baseline tend to be what are called exceptional responders (i.e., complete or partial response lasting more than 12 months) when treated with sotorasib [177]. Based on that clinical efficacy and safety profile, sotorasib was therefore recently approved by the European Commission, which granted it a conditional marketing authorization, underlining that confirmatory trials are awaited.

### 9.3. Adagrasib (MRTX849): Pre-Clinical and Clinical Development

The second small-molecule inhibitor of *KRAS* c.34G > T (p.G12C) that is under evaluation in clinical trials is adagrasib (formerly MRTX849) [178]. Similar to sotorasib, adagrasib impairs cell viability in pancreatic and lung AC monolayer and spheroid models harboring *KRAS* c.34G > T (p.G12C) [179]. Beyond *KRAS* c.34G > T (p.G12C), no single genomic co-alteration predicted the antitumor activity of adagrasib. A preliminary report on the clinical efficacy of adagrasib confirmed an objective response and a disease control rate in 45% and 96% of evaluable patients in the NSCLC cohort, respectively (*n* = 79) [178].

More recently, results from the KRYSTAL-1 phase II study enrolling 116 patients with NSCLC harboring a *KRAS* c.34G > T (p.G12C) mutation who received adagrasib following prior treatment with immunotherapy and chemotherapy were presented by Spira et al. at the 2022 ASCO annual meeting [179]. Adagrasib showed promising efficacy, with ORR of 42.9% and the DCR of 79.5%. Median DOR was 8.5 months, median PFS was 6.5 months and median OS was 12.6 months.

The most common side effects reported on adagrasib are those of gastrointestinal pertinence (nausea, vomiting and diarrhea), fatigue and asthenia, a pattern similar to that observed after sotorasib administration. Discrete hepatic functional impairment, associated with alanine aminotransferase increase, was also registered among patients treated with adagrasib. Concomitant mutations like in *STK11* gene did not exclude adagrasib efficacy as reported, regardless of limited subgroup numerosity [180,181]. Adagrasib is the second-most selective KRAS inhibitor currently being reviewed by the FDA for accelerated approval for patients with lung cancer who harbor the *KRAS* c.34G > T (p.G12C) mutation. A phase III trial (KRYSTAL-12) evaluating adagrasib monotherapy versus docetaxel in pretreated patients with *KRAS* c.34G > T (p.G12C)-mutated NSCLC is ongoing (ClinicalTrials.gov NCT04685135).

### 9.4. Other KRAS-Targeted Molecules

Several other direct *KRAS* c.34G > T (p.G12C) allosteric inhibitors are being tested in the clinical setting as monotherapy and in combination with other therapies. These *KRAS* c.34G > T (p.G12C) inhibitors include GDC-6036 (phase I; ClinicalTrials.gov NCT04449874), D-1553 (phase I/II; ClinicalTrials.gov NCT04585035) JDQ443 (phase I/II; ClinicalTrials.gov. NCT04699188), JAB-21822 (phase I/II: ClinicalTrials.gov NCT 05002270), and LY3499446, and ARS-3248/JNJ-74699157, which are next-generation ARS-1620 inhibitors (ClinicalTrials.gov NCT04006301, and ClinicalTrials.gov NCT04585035) with inhibitory activity at least 10 times higher than that of sotorasib and adagrasib (ClinicalTrials.gov NCT04006301).

Compared with direct inhibition, degradation could be a more potent strategy that affects cell proliferation and downstream signaling responses [182,183], which is why recently, proteolysis-targeting chimaeras (PROTACs) targeting KRAS c.34G > T (p.G12C) proteins are being explored.

Based on covalent inhibitors targeting *KRAS* c.34G > T (p.G12C), PROTACs are designed to target *KRAS* c.34G > T (p.G12C). Trials based on PROTAC molecules demonstrated the first clinical proof-of-concept against two of the most relevant cancer targets: the estrogen receptor (ER) and androgen receptor (AR). Clinical trials of oral PROTACs (ARV-110 and ARV-471) have manifested uplifting outcomes for prostate and breast cancer treatment [184,185]. Encouraged by such results, the targeted protein degradation field (TPD) is now assured to reach undrugged targets.

Finally, a nanoparticle-formulated mRNA based cancer vaccine (V941) that targets four of the most prevalent occurring *KRAS* mutations (c.35G > A, p.G12D; c.35G > T, p.G12V; c.38G > A, p.G13D and c.34G > T, p.G12C) is in developing stages as monotherapy or in combination with pembrolizumab in NSCLC, CRC or PDAC patients (ClinicalTrials.gov NCT03948763).

## 10. Biological Mechanism of Acquired Resistance

### 10.1. Acquired Resistance to Sotorasib and Adagrasib

Biological mechanisms of acquired resistance to sotorasib and adagrasib have been described in both pre-clinical and clinical experiences. Similar to what is known concerning other oncogenic drivers in NSCLC [186], new-onset *KRAS*-activating mutations and *KRAS* amplification can mediate acquired resistance through RAS signaling pathway activation [187]. With the development of KRAS inhibitors, some findings highlighting the diversity of *KRAS* mutations emerged in response to their use, thereby limiting the potential development of efficient next-generation *KRAS* c.34G > T (p.G12C) inhibitors [188,189,190]. Some of these mutations display differential sensitivity to either sotorasib or adagrasib, thus providing a theoretical rationale for a sequencing strategy [188,189,190].

In the preclinical and clinical setting, presumed mechanisms of acquired resistance were described in 45% of cases for adagrasib and 60% of cases for sotorasib. These observations provide a rationale for combined therapy to prevent or delay acquired resistance [177,190]. These resistance mechanisms can involve on-target or bypass alterations, which lead to the abnormal activation of downstream or connecting signaling pathways as well as phenotypic transformation mechanisms to either SCLC or SCC [188,189,190].

### 10.2. Acquired Resistance Mechanisms to Either Sotorasib or Adagrasib

Mutations in *KRAS* that confer acquired resistance to *KRAS* c.34G > T (p.G12C) inhibitors have been described in vitro and in the clinical setting [188,189,190] and occur primarily in switch-II binding pocket, thereby altering drug binding [187].

Patients who progressed on adagrasib have shown secondary mutations that alter drug binding [190]. In this subgroup of patients, the secondary mutations were expressed as double mutant alleles in Ba/F3 cell lines in order to shape their mechanistic impacts on *KRAS* c34G > T (p.G12C) inhibitor sensitivity. Awad et al. [190] described that Switch-II binding pocket mutations, including c.204G > T (p.R68S) and codons H95 and Y96, conferred resistance to Adagrasib in Ba/F3 cells, in contrast with the control *KRAS* c.34G > T (p.G12C) allele. According to their findings, the Y96C mutation caused cross-resistance to both adagrasib and sotorasib in vitro. In sotorasib-resistant clones, the *KRAS* c.38G > A (p.G13D) variant was the most frequent secondary mutation (i.e., 23% of *KRAS* secondary mutations), followed by c.203G > T (p.R68M) and c.175G > T (p.A59S) (21% and 2%, respectively). In adagrasib-resistant clones, *KRAS* Q99L was the most frequent secondary mutation (i.e., 52.8% of *KRAS* secondary mutations), followed by codon Y96 and c.204G > T (p.R68S) (i.e., 15, 3% and 13.9%, respectively) [186].

Tanaka et al. described how RM-018 (a novel *KRAS* c.34G > T (p.G12C) inhibitor that exploits cyclophilin A to bind and inhibit *KRAS* c.34G > T (p.G12C) in its GDP bound state) could overcome *KRAS* codon Y96 resistance mutation [189].

Other mutations were involved in in vitro resistance to MRTX1257 (a compound highly related to adagrasib and sotorasib) through the inhibition of GTP hydrolysis or the increase of GDP to GTP nucleotide exchange (like mutations occurring at codons 13, 59, 61, 117 and 146) [188,190,191,192,193,194].

The abovementioned in vitro experiments highlight the different sensitivities to *KRAS* c.34G > T (p.G12C) inhibitors for different acquired mutations. Some mutations like G13D and A59S remained partially sensitive to adagrasib while they conferred strong resistance to sotorasib [186]. In the same way, codon Q99 and H95 secondary mutations, which conferred resistance to adagrasib, remained sensitive to sotorasib [188,190]. In the plethora of shared mutations between sotorasib and adagrasib, only the ones in codon Y96 proved to confer strong cross-resistance to both *KRAS* c.34G > T (p.G12C) inhibitors [188]. The trials in clinical setting highlighted the main mechanisms of acquired resistance to sotorasib and adagrasib through secondary mutations within the drug-binding pocket and activating mutations in *KRAS*. Up to 53% of patients who showed any putative mechanisms of resistance to adagrasib had at least one acquired *KRAS* mutation or *KRAS* amplification. Acquired mutations within the switch II binding pocket– Y96 codon, c.204G > T (p.R68S), H95 codon, likewise mutations in *KRAS* c.35G > A (p.G12D) and c.35G > T (p.G12V) were reported. A single patient showed a high level of *KRAS* c.34G > T (p.G12C) amplification at relapse.

Regarding other alterations associated with resistance to *KRAS* c.34G > T (p.G12C) inhibitors, among 32 NSCLC patients enrolled in Codebreak 100 and Codebreak 101, acquired resistance was identified in 78% of cases (including alterations in *KRAS*, *NRAS*, *BRAF*, *EGFR*, *FGFR2*, *MYC* and other genes). Among them, four patients had pG12D, pG12V, pG12F and pV8L changes, while three patients had *KRAS* amplifications [177]. The c.35G > T (p.G12V) mutation in this report was shown to decrease the antiproliferative effects of both sotorasib and adagrasib.

Sotorasib has been described as improving the therapeutic efficacy of other targeted agents, i.e., with HER kinase inhibitors (i.e afatinib), SHP2 inhibitors (RMC-4550) and MEK inhibitors (trametinib), and in this setting, several studies are ongoing (Table 6 and Table 7). This might emerge as a potential strategy to overcome acquired resistance to these compounds [166]. A synergic activity of adagrasib was found when this drug was associated with other targeted agents (afatanib or RMC 4550) in xenograph models of NSCLC. Other efficient mechanisms that might enhance adagrasib efficacy have been described in vivo, like targeting downstream KRAS effectors such as mTOR and cyclid D family [180].

## 11. Activation of RTKs and RAS Downstream Signaling Pathways

One of the main mechanisms of acquired resistance is the activation of bypass signaling pathways described in oncogene-driven NSCLC, whether these are vertical and/or parallel pathways [186]. Targeting these pathways in addition to direct KRAS inhibition may be useful for overcoming both ab initio and acquired resistance mechanisms, and may result in more durable responses as well.

Tumoral heterogeneity of resistant tumors is explained partially by the co-occurrence of numerous mechanisms of acquired resistance to adagrasib [189], including mutations in *NRAS* (i.e., c.182A > T, p.Q61L; c.181C > A, p.Q61K; and c.182A > G, p.Q61R), *BRAF* c.1799T > A (p.V600E) and *MAP2K1* (i.e., c.171G > C, p.K57N and c.167A > C, p.Q56P) genes [189]. In particular, it has been proposed that the polyclonal RAS-MAPK reactivation is a central resistance mechanism, specifying that *NRAS* c.181C > A (p.Q61K) not only was mechanistically confirmed as a mutation on adagrasib resistance but was also found to confer resistance to sotorasib in vitro as well [177]. Other mutations or amplifications detected in five NSCLC patients with progressive disease on adagrasib were found in *RET*, *PIK3CA* and *MET* as defined as well as acquired bypass mechanisms of resistance. Moreover, other alterations in the clinical setting were described in the RAS–MAPK pathway as acquiring resistance to sotorasib like in *NRAS*, *BRAF*, *EGFR*, *FGFR2* and *MET*, expressed as mutations or amplifications of these genes [177,190].

In order to overcome acquired resistance to *KRAS* c.34G > T (p.G12C), several promising strategies are being elaborated, like targeting the specific RTK involved in bypassing the signaling pathways either directly or by inhibiting upstream and downstream effectors of *KRAS* c34G > T(G12C). In this perspective, different strategies are currently being proposed [191,192,193,194,195,196]. The combination of sotorasib with crizotinib has proved to be more efficient in inhibiting ERK, AKT and MET activation in H23 sotorasib-resistant cells that acquired *MET* amplification, compared to monotherapy. Likewise, this combination induced apoptosis and notably decreased tumor growth in *MET*-amplified H23 sotorasib-resistant xenograft models. In the same direction, the combination of sotorasib and capmatinib (a specific MET inhibitor) decreased cell viability [195].

Even though these results that support the further targeting of downstream or upstream effectors of *KRAS* c.G34 > T (pG12C) might seem to suggest an auspicious strategy, it might not be sufficient to bypass resistance. As a consequence, several clinical trials in development stages are evaluating the efficacy of *KRAS* c.34G > T (p.G12C) inhibitors in combination with chemotherapy [197,198] or with targeted therapies like cetuximab, afatinib, pembrolizumab and SHP2, mTOR and CDK4/6 inhibitors [178,197,199,200] (Table 8).

CDK4/6 are key regulators of the cell cycle and can be affected by KRAS through multiple signaling pathways, such as the RAF-MEK-ERK and PI3K/AKT pathways. Hallin et al. showed that in several MRTX849-refractory models, the combination of MRTX849 and palbociclib led to significant general tumor regression, which may be due to the increased inhibition of the retinoblastoma protein/E2F transcription factor pathway [180]. Trials evaluating the combination of CDK4/6 inhibitors with *KRAS* c.34G > T (p.G12C) selective inhibitors are ongoing [180] (Table 8).

Another promising strategy for overcoming resistance might be the inhibition of aurora kinase A (AURKA), an enzyme implicated in cell division. ARS-1620 (a potent, orally bioavailable covalent inhibitor of *KRAS* c.34G > T (p.G12C), when combined with alisertib (AURKA inhibitor), was observed to have a significant synergistic effect in ARS-1620-refractory *KRAS* c.34G > T (p.G12C) mutant cancer. The mechanistic inhibition of AURKA has been shown not only to impair normal cell division and proliferation but also to affect KRAS–c RAF interaction, leading to a challenging activation of RAF [201].

Lastly, it appears that the PI3K-AKT pathway is less dependent on the *RAS* pathway and less sensitive to *KRAS* c.34G > T (p.G12C) inhibitors; nevertheless, enhanced antitumor activity was seen when adagrasib was combined with afatinib, RMC-550, vistusertib (mTOR) and palbociclib in xenograft models [180]. Combined strategies targeting upstream effectors (i.e., SOS1 or SHP2) and the PI3KAKT- mTOR pathway could be relevant in the near future.

## 12. MEK Inhibition and Therapeutic Strategies to Overcome on-Target Mechanisms of Acquired Resistance to *KRAS* c.34G > T (p.G12C) Inhibitors

The combination of sotorasib with inhibitors of HER kinase, EGFR, SHP2, PI3K/AKT and MEK was evaluated in vitro, and the results of a CodeBreak 101 study (NCT04185883) favored the combination of MEK inhibitors (trametinib) with sotorasib versus one single agent. The addition of a MEK inhibitor to docetaxel demonstrated better outcomes in patients with *KRAS*-mutant NSCLC compared with either single-agent treatment, combination with erlotinib or docetaxel alone [202,203]. Different combinatorial strategies with a *KRAS* c.34G > T (p.G12C) inhibitor and/or MEK with chemotherapy are being evaluated in in vitro and in the clinical setting [204,205]. MEK inhibitors might constitute an alternative strategy in MET-amplified tumors, as preclinical experiences highlighted the importance of the MAPK pathway in this setting [206] (Table 7). The MEK inhibitor selumetinib showed interesting activity in vitro when associated with sotorasib as assessed by the complete inhibition of ERK phosphorylation in H23 MET-amplified sotorasib-resistant cells [195]. In spite of the preclinical efficacy of these therapeutic strategies, expanding on these results in dedicated clinical trials will prove complex due to the small proportion of patients who present these alterations.

In a phase 1b study, the addition of the oral MEK inhibitor binimetinib to the treatment combination of cisplatin and pemetrexed as first-line therapy in advanced KRAS-mutated metastatic NSCLC showed no signal of anti-tumor activity in terms of PFS or OS. (Froesch et al., 2021). Novel strategies focused on KRAS scaffold targets, namely SOS1 and SHP2, are being actively investigated to overcome acquired resistance to KRAS c.34G > T (p.G12C) inhibitors. SHP2 is a tyrosine phosphatase that is required for complete RAS-MAPK activation, and it is essential for KRAS-mutant carcinogenesis [207].

Multiple SHP2 inhibitors are being investigated both as monotherapy and in combination with KRAS c.34G > T (p.G12C) inhibitors like RMC 4630, RMC 4550, RLY 1971, TNO15 [180,201,208,209,210,211], (clinicaltrials.gov NCT 04252339, NCT03114319, NCT04699188). A novel inhibition strategy into *KRAS* c.34G > T (p.G12C) therapeutic approaches has yielded a new target in the SOS1 catalytic domain that affects the EGFR/MAPK pathway. Through the inhibition of SOS1, the binding of KRAS to GTP and its constitutively active state is reduced. BI-3046 and BI 1,701,963 are orally bioavailable small-molecule SOS1 inhibitors, that bind to the catalytic domain of SOS1, thereby preventing the interaction with KRAS. They showed effectiveness in KRAS-driven cancers when combined with MEK inhibitors (trametinib) [212].

Further on, the identification of BI-3406 in pre-clinical studies (a highly potent selective small-molecule SOS1 inhibitor that binds to the catalytic domain of SOS1 thereby prevents the interaction with KRAS) allowed for broadening the spectrum of activity by demonstrating that the therapeutic efficacy of this molecule was not limited only to *KRAS* c.34G > T (p.G12C) mutant models but also showed activity in *KRAS* c.35G > T (p.G12V) (SW620cells), *KRAS* c.38G > A (p.G13D) (LoVo cells) and *KRAS* c.34G > A (p.G12S) (A549 cells) as well by inducing tumor growth inhibition in xenograph models [213]. Moreover, BI-3406 was found to prevent ERK activation rebound following sotorasib treatment in vitro [213].

Importantly, BI-3406 demonstrated promising pre-clinical results in overcoming acquired resistance to *KRAS* c.34G > T (p.G12C) inhibitors when combined with treatments that target downstream *KRAS* c.34G > T (p.G12C) effectors. BI-3406 attenuates the feedback reactivation induced by MEK inhibitors enhancing as a consequence the sensitivity of KRAS-dependent cancers to MEK inhibition. Combined SOS1 and MEK inhibition represents a novel and effective therapeutic concept for addressing KRAS-driven tumors. Finally, the combination of sotorasib and BI-3406 decreased RAS-GTP levels and Inhibit ERK phosphorylation, while it affected less the AKT activation in *MET*-amplified sotorasib resistant H23 cells [195].

## 13. Lineage Plasticity and Acquisition of Features of Epithelial-To-Mesenchymal Transition as a Mechanisms of Acquired Resistance

Despite the genetic and histologic transformation to either SCLC or SCC representing a well-known biological mechanism of acquired resistance to targeted therapies in NSCLC [192,193], the evidence of trans-differentiation to SCLC in the context of acquired resistance to *KRAS* c.34G > T (p.G12C)-targeted therapies is scarce and limited to clinical case reports [190]. Other co-mutations like *STK11* deletion seem to drive lineage transformation to SCC in *KRAS* mutated lung AC [214].

Another particular and praiseworthy resistance mechanism is represented by the epithelial-to-mesenchymal transition (EMT). The ability of acquisition of EMT features in in vitro and in xenograph cells is interpreted as a potential acquired resistance mechanism to sotorasib in the setting of *KRAS* c.34G > T (p.G12C) mutant NSCLC [196].

In vitro efficacy was demonstrated by Adachi et al. when IGF1R and FGFR1 were targeted: RTKs that intermediated acquired resistance to *KRAS* c34 G > T (p.G12C) inhibitors through EMT [196]. The researchers reported as well that EMT might be the cause of both intrinsic and acquired resistance by activating the PI3K pathway in the presence of the KRAS G12C inhibitor.

The triple combination of sotorasib, linsitinib (IGF1R inhibitor) and infigratinib (FGFR1 inhibitor), in contrast with doublet therapies (sotorasib with either linsitinib or infigatinib), showed efficacy by decreasing the cell proliferation and phosphorylation of AKT, ERK and S6 in H358-sotorasib-resistant cells [196]. Furthermore, this triple combination was shown to be the only treatment that reversed acquired resistance to sotorasib in vivo through the successful suppression of PI3K-AKT, MAPK and S6 phosphorylation [196].

## 14. Immune Check Point Inhibitors in *KRAS*-Mutant NSCLC, Current and Future Therapeutic Prospectives

As previously mentioned, *KRAS*-mutant NSCLC patients are expected to benefit from immune checkpoint inhibitors (ICIs) [136], and this has been demonstrated in particular in *KRAS* c.34G > T (p.G12C) mutant individuals [137,138].

Patients with *KRAS* c.34G > T (p.G12C) aberrations were included in frontline immunotherapy clinical trials including KEYNOTE-189, KEYNOTE-042, KEYNOTE-024 and CheckMate-227. An exploratory analysis was performed in KEYNOTE-042 aiming to assess the prevalence of *KRAS* mutations and their associations with ICI efficacy. Findings from this exploratory analysis showed that pembrolizumab monotherapy was associated with a 58% reduced risk of death (HR = 0.42 [95% CI, 0.22–0.81]) in patients with any *KRAS* mutation and a 72% (HR = 0.28 [95% CI, 0.09–0.86]) reduced risk in patients with the *KRAS* c.34G > T (p.G12C) mutation compared with chemotherapy [215].

The benefit from immunotherapy was limited by concomitant mutations like *STK11* [174].

Sotorasib and adagrasib have shown enhanced antitumor efficacy when combined with anti-PD-1/PD-L1 therapy. Clinical trials are under development to investigate this interaction [216]. In genetically engineered mouse models with *KRAS* c.34G > T (p.G12C), the combination of adagrasib with an anti-PD-1 resulted in a complete and prolonged response in vivo and increased PFS compared with single-agent monotherapy [217].

Currently, multiple trials are investigating the potential of *KRAS* c.34G > T (p.G12C) inhibitors with existing immune checkpoint inhibitors (Table 9).

## 15. Co-Occurring Mutations Might Alter Clinical Benefit of ICIs

In preclinical studies, *KRAS* c.34G > T (p.G12C) direct inhibitors were noted to upregulate a pro-inflammatory tumor microenvironment and enhance anti-tumor T cell activity [166].

*KRAS/STK11* aberrations, on the other hand, promote a so-called ‘cold tumor’ immune microenvironment, generally with limited PD-L1 expression and reduced infiltrating CD8+ T cells [218]. While *TP53* co-mutation seems to be associated with a better response to ICIs, *STK11* and *KEAP1* co-mutations seem to impair response to ICIs

Indeed, *TP53/KRAS* co-mutated patients show higher PD-L1 expression, increased count of CD8 + TILs [141] and a higher mutational load compared with either *STK11* or *KEAP1* co-mutated patients; these findings may explain the better PFS in favor of patients harboring *TP53*/*KRAS* mutations treated with anti-PD-1 therapy compared with patients without mutations in *TP53* and *KRAS* genes [141]. An exception is *KRAS* c.35G > A (p.G12D) mutation combined with *TP53* mutations, which predict a lower TMB and as a consequence, less efficacy of immunotherapies in lung AC [219,220].

As aforementioned, *STK11* and *KEAP1* co-occurring mutations have been associated with ICI resistance in *KRAS*-mutant patients, independently of PD-L1 expression. Skoulidis et al. showed *STK11*/*LKB1* alterations as a major driver of primary resistance to PD-1 blockade in *KRAS*-mutant patients, identifying *STK11*/*LKB1* alterations as the most prevalent genomic driver of primary resistance to PD-1/PD-L1 inhibitors in *KRAS*-mutant NSCLC [87,174,219]

Patients harboring *KRAS*, *STK11* and/or *KEAP 1* co-mutations treated with car-boplatin-paclitaxel combined with either atezolizumab, bevacizumab or atezolizumab and bevacizumab had a worse outcome in terms of PFS and OS compared with *STK11* and *KEAP1* wt patients as showed in the post hoc analysis of the IMpower 150 trial [219].

Recently, Feng et al. reported a positive association between PD-L1 expression (i.e., ≥50%) and levels of stromal SHP2 (*p* = 0.039) [220]. In the same study, SHP2 was also found to promote a pro-inflammatory microenvironment through a higher expression of CD8+/CD4+ T-cells and CD64+ macrophages found in the stromal compartment of patients with high stromal SHP2 expression. Finally, the authors reported that despite small subgroup sizes, our subgroups analyses showed that patients with high SHP2 expression and PD-L1 ≥ 1% had significantly prolonged PFS and OS [220,221]. Based on the above-described preclinical results, several clinical trials are currently ongoing evaluating the association of anti-PD-1/PD-L1 agents with either sotorasib or adagrasib (Table 9).

The presence of the SWI/SNF related, matrix-associated, actin-dependent regulator of chromatin, subfamily A, member 4 (*SMARCA4*) are co-mutations in *KRAS*-mutated NSCLC (nearly 10% of cases) and are associated with poor immunotherapy outcomes [222,223,224,225]. This poor response could be associated with the influence of SMARCA4 alterations that decrease the production of the SMARCA4 protein, which is involved in chromatin remodeling and it is important in regulating the binding of transcription factors to DNA [226]. However, further studies are required in this setting.

## 16. Conclusions

Although neglected until few years ago, the clinical role of *KRAS* mutations has now gained a central role in the oncologic management of lung cancer patients. In recent years, it has become evident that the occurrence of *KRAS* mutations in lung cancer patients may reflect an opportunity for new drug inhibition, essentially with those directly targeting the G12C variant. Sotorasib is currently approved in this context, but other molecules are now under investigation in several clinical trials. Mechanisms of primary and acquired resistance, including mutations in *KRAS* as well as in other genes not belonging to the same MAP kinase pathway, may lead to new, innovative therapeutic options, with other TKI but including ICI as well, thus significantly prolonging patients’ survival.

## Figures and Tables

**Table 1 cancers-14-04103-t001:** Cancers with the highest *KRAS* mutations frequencies.

Cancer	*KRAS* Mutations Percentage
PDAC	88%
CRC	45–50%
Lung AC	30–35%

AC: AdenoCarcinoma; CRC: ColoRectal Cancer; PDAC: Pancreatic Ductal AdenoCarcinoma (AACR Project GENIE Consortium, 2017 [7]; Prior et al., 2020 [6]; Uras et al., 2020 [8]; Reck et al., 2021 [5]).

**Table 2 cancers-14-04103-t002:** Rates of *KRAS* mutations in NSCLC and subtypes.

Cancer	Mutation Rates			
	KRAS	KRAS Exon 2 Mutations Rate	KRAS Exon 3 Mutations Rate	KRAS Exon 4 Rate Considering the Total of KRAS Mutations
NSCLC	25–30%	34%	1.89%	0.24%
Lung AC	30–35%	27%	7%	<1%
SCC	3–4%	7%	<3%	<0.5%

AC: AdenoCarcinoma; NSCLC: Non-Small Cell Lung Cancer; SCC: Squamous-Cell Carcinoma (AACR Project GENIE Consortium, 2017 [7]).

**Table 3 cancers-14-04103-t003:** Rates of *KRAS* exons 2, 3 and 4 mutations in NSCLC.

KRAS Mutations	Frequencies
**Exon 2 codon 12**	
c.34G > T (p.G12C)	40%
c.35G > T (p.G12V)	19–21%
c.35G > A (p.G12D)	15–17%
c.35G > C (p.G12A)	6%
c.34G > A (p.G12S)	4%
c.34G > C (p.G12R)	4%
**Exon 2 codon 13**	
c.37G > T (p.G13C)	6.8%
c.38G > A (p.G13D)	0.8%
c.37G > C (p.G13R)	0.6%
c.37G > A (p.G13S)	NR
**Exon 3 codon 59**	
c.176C > A (p.A59E)	NR
c.176C > G (p.A59G)	NR
c.175G > A (p.A59T)	NR
**Exon 3 codon 61**	NR
c.183A > C (p.Q61H)	1.2%
c.182A > T (p.Q61L)	0.4%
c.182A > G (p.Q61R)	0.06%
c.180_181delinsAA (p.Q61K)	0.02%
c.181C > G (p.Q61E)	NR
**Exon 4 codon 146**	
c.436G > C (p.A146P)	NR
c.436G > A (p.A146T)	NR
c.437C > T (p.A146V)	NR

NR: Not Relevant; NSCLC: Non-Small Cell Lung Cancer (AACR Project GENIE Consortium, 2017 [7]).

**Table 4 cancers-14-04103-t004:** Rates of *KRAS* mutations in lung AC subdivided in different ethnicities.

	KRAS Mutation Rates			
	Caucasians	Asians	Africans	Americans
**KRAS exon2**				
c.34G > T (p.G12C)	34%	24.5%	NA	41.2%
c.35G > T (p.G12V)	21%	1.5%	NA	19%
c.35G > A (p.G12D)	14%	25.5%	28.5%	15%
c.35G > C (p.G12A)	9.5%	1.3%	NA	7%
c.34G > A (p.G12S)	1%	0.6%	NA	1%
c.34G > C (p.G12R)	1%	0.6%	NA	1%
c.37G > T (p.G13C)	4%	0.7%	NA	7%
c.38G > A (p.G13D)	1–2%	0.7%	42.8%	1%
c.37G > C (p.G13R)	0.5%	0.5%	NA	0.5%
c.37G > A (p.G13S)	NR	NR	NR	NR
**KRAS exon3**				
c.183A > C (p.Q61H)	1%	0.8%	NA	4%
c.182A > T (p.Q61L)	0.5%	0.4%	NA	1%
c.182A > G (p.Q61R)	0.06%	0.05%	NA	0.05
c.180_181delinsAA (p.Q61K)	0.02%	NR	NA	0.02
c.181C > G (p.Q61E)	NR	NR	NA	NR
c.176C > A (p.A59E)	NR	NR	NA	NR
c.176C > G (p.A59G)	NR	NR	NA	NR
c.175G > A (p.A59T)	NR	NR	NA	NR
**KRAS exon4**	NR	NR	NA	NR
c.436G > C (p.A146P)	NR	NR	NA	NR
c.436G > A (p.A146T)	NR	NR	NA	NR
c.437C > T (p.A146V)	NR	NR	NA	NR

AC, AdenoCarcinoma; NA: Not Available; NR: Not Relevant (Boch et al., 2013 [19]; Elghissassi et al., 2014 [31]; Kim et al., 2014 [32]; Li et al., 2014 [33]; Nadal et al., 2014 [34]; Arrieta and Ramírez-Tirado et al., 2015 [35]; Dhieb et al., 2019 [36]; Judd et al., 2021 [18]).

**Table 5 cancers-14-04103-t005:** Direct KRAS inhibition, ongoing studies.

Title	Status	Condition	Intervention	NCT Number
A Study of Sotorasib (AMG 510) in Participants With Stage IV NSCLC Whose Tumors Harbor a KRAS p.G12C Mutation in Need of First-line Treatment (CodeBreaK201)	Recruiting	- NSCLC	- Sotorasib	NCT04933695
FIH Study of JAB-21822 in Adult Patients With Advanced Solid Tumors Harboring KRAS G12C Mutation in China	Recruiting	- NSCLC - Colorectal cancer - Solid Tumor	- JAB-21822	NCT05009329
Testing the Use of Targeted Treatment (AMG 510) for KRAS G12C Mutated Advanced Non-squamous Non-small Cell Lung Cancer (A Lung-MAP Treatment Trial)	Recruiting	- Recurrent and metastatic Non-Squamous NSCLC	- Sotorasib	NCT04625647
Sotorasib in Advanced KRASG12C-mutated Non-small Cell Lung Cancer Patients With Comorbidities (SOLUCOM)	Recruiting	- Recurrent or metastatic NSCLC - Mutation cancer	- Sotorasib	NCT05311709
Study to Evaluate D-1553 in Subjects With Solid Tumors	Recruiting	- Solid Tumor, - NSCLC - CRC	- D-1553	NCT04585035

CRC: ColoRectal Cancer; NSCLC: Non-Small Cell Lung Cancer.

**Table 6 cancers-14-04103-t006:** KRAS scaffold inhibition (SHP2/SOS1), ongoing studies.

Title	Status	Condition	Intervention	NCT Number
SHP2 Inhibitor BBP-398 in Combination With Nivolumab in Patients With Advanced Non-Small Cell Lung Cancer With a KRAS Mutation	Recruiting	- NSCLC - Solid Tumor	- BBP-398 with nivolumab	NCT05375084
Combination Therapy of RMC-4630 and LY3214996 in Metastatic KRAS-mutant Cancers (SHERPA)	Recruiting	- Pancreatic Cancer - CRC - NSCLC - KRAS Mutation-Related Tumors	- RMC-4630 - LY3214996	NCT04916236
Combination Study of RMC-4630 and Sotorasib for NSCLC Subjects With KRASG12C Mutation After Failure of Prior Standard Therapies	Recruiting	- NSCLC	- RMC-4630 - Sotorasib	NCT05054725
Dose-Escalation/Expansion of RMC-4630 and Cobimetinib in Relapsed/Refractory Solid Tumors and RMC-4630 and Osimertinib in EGFR Positive Locally Advanced/Metastatic NSCLC	Active, not recruiting	- Solid tumor	- RMC-4630 - Cobimetinib - Osimertinib	NCT03989115
A Phase 1 Study of the SHP2 Inhibitor BBP-398 in Combination With Nivolumab in Patients With Advanced Non-Small Cell Lung Cancer With a KRAS Mutation	Recruiting	- NSCLC - Solid Tumor	- BBP-398 with Nivolumab	NCT05375084
A Dose Escalation Study of SHP2 Inhibitor in Patients With Solid Tumors Harboring KRAS of EGFR Mutations	Recruiting	- NSCLC - CRC - Pancreatic Cancer - Solid Tumor	- HBI-2376	NCT05163028
Adagrasib in Combination With TNO155 in Patients With Cancer (KRYSTAL 2)	Active, not recruiting	- Advanced Cancer - Metastatic Cancer - Malignant Neoplastic Disease	- MRTX849 - TNO155	NCT04330664
Study of JDQ443 in Patients With Advanced Solid Tumors Harboring the KRAS G12C Mutation (KontRASt-01)	Recruiting	- KRAS G12C Mutant Solid Tumors - NSCLC - CRC	- JDQ443 - TNO155 - Tislelizumab	NCT04699188

AC: AdenoCarcinoma; CRC: ColoRectal Cancer; NSCLC: Non-Small Cell Lung Cancer.

**Table 7 cancers-14-04103-t007:** Combination strategies of KRAS inhibition and upstream/downstream pathways (EGFR/RAS/RAF/MEK/HER2), ongoing studies.

Title	Status	Condition	Intervention	NCT Number
Assess Efficacy & Safety of Selumetinib in Combination With Docetaxel in Patients Receiving 2nd Line Treatment for v-Ki-ras2 Kirsten Rat Sarcoma Viral Oncogene Homolog (KRAS) Positive NSCLC (SELECT-1)	Active, not recruiting	- Locally Advanced or Metastatic - NSCLC	- Selumetinib - Docetaxel - Placebo - Pegylated G-CSF	NCT01933932
Trametinib and Docetaxel in Treating Patients With Recurrent or Stage IV KRAS Mutation Positive Non-small Cell Lung Cancer	Active, not recruiting	- KRAS Gene Mutation - Recurrent ore metastatic NSCLC	- Docetaxel - Laboratory Biomarker Analysis - Trametinib	NCT02642042
Rigosertib Plus Nivolumab for KRAS+ NSCLC Patients WhoProgressed on First-Line Treatment	Recruiting	- Metastatic NSCLC - AC	- Rigosertib - Nivolumab	NCT04263090
Binimetinib and Hydroxychloroquine in Patients With Advanced KRAS-mutant Non-Small Cell Lung Cancer	Recruiting	- NSCLC - KRAS Mutation-Related Tumors	- Binimetinib - Hydroxychloroquine	NCT04735068
MEK Inhibitor Combined With Anlotinib in the Treatment of KRAS-mutated Advanced Non-small Cell Lung Cancer	Recruiting	- NSCLC	- Trametinib - Anlotinib	NCT04967079
Regorafenib and Methotrexate in Treating Participants With Recurrent or Metastatic KRAS Mutated Non-Small Cell Lung Cancer	Active, not recruiting	- KRAS Gene Mutation - Metastatic Malignant Neoplasm in the Brain - Recurrent or metastatic NSCLC	- Methotrexate - Pharmacokinetic Study - Regorafenib	NCT03520842
A Phase I/IB Trial of MEK162 in Combination With Erlotinib in NSCLC Harboring KRAS or EGFR Mutation	Active, not recruiting	- Lung cancer - NSCLC	- MEK162 - Erlotinib	NCT01859026
Trametinib and Pembrolizumab in Treating Patients With recurrent NSCLC that is metastatic, unresectable or locally advanced.	Active, not recruiting	- Unreseactable or Metastatic NSCLC	- Pembrolizumab - Trametinib - Pharmacokinetic study	NCT03225664
Dose-Escalation/Expansion of RMC-4630 and Cobimetinib in Relapsed/Refractory Solid Tumors and RMC-4630 and Osimertinib in EGFR Positive Locally Advanced/Metastatic NSCLC	Active, not recruiting	- Solid tumor	- RMC-4630 - Cobimetinib - Osimertinib	NCT03989115
JAB-21822 Activity in Adult Patients With Advanced Solid Tumors Harboring KRAS G12C Mutation	Recruiting	- Advanced Solid Tumor - NSCLC - CRC	- JAB-21822 (KRAS G12C inhibitor) - Cetuximab (EGFR inhibitor)	NCT05002270
Pembrolizumab and Trametinib in Treating Patients With Stage IV Non-Small Cell Lung Cancer and KRAS Gene Mutations.	Active, not recruiting	- KRAS Gene Mutation - Recurrent or Metastatic Non-Squamous NSCLC.	- Pembrolizumab - Trametinib	NCT03299088
Tarlox and Sotorasib in Patients With KRAS G12C Mutations	Recruiting	- NSCLC	- Sotorasib and Tarloxotinib	NCT05313009
Phase 1/2 Study of VS-6766 + Sotorasib in G12C NSCLC Patients (RAMP203)	Recruiting	- NSCLC - KRAS Activating Mutation	- VS-6766 and sotorasib	NCT05074810
A Study of VS-6766 v. VS-6766 + Defactinib in Recurrent G12V, Other KRAS and BRAF Non-Small Cell Lung Cancer (RAMP202)	Recruting	- NSCLC - KRAS Activating Mutation	- VS-6766 •Drug: VS-6766 and Defactinib	NCT04620330
A Study to Evaluate the Safety, Pharmacokinetics, and Activity of GDC-6036 Alone or in Combination in Participants With Advanced or Metastatic Solid Tumors With a KRAS G12C Mutation	Recruiting	- NSCLC - CRC - Advanced Solid Tumors	- GDC-6036 - Atezolizumab - Cetuximab - Bevacizumab - Erlotinib - GDC-1971 - Inavolisib	NCT04449874
Phase 1b/2 Study of Futibatinib in Combination With Binimetinib in Patients With Advanced KRAS-mutant Cancer	Recruiting	- Advanced or Metastatic Solid Tumors Irrespective of Gene Alterations - NSCLC - KRAS Gene Mutation	- Futibatinib and Binimetinib	NCT04965818
Phase 1/2 Study of MRTX849 in Patients With Cancer Having a KRAS G12C Mutation KRYSTAL-1	Recruiting	- Advanced Cancer - Metastatic Cancer - Malignant Neoplastic Disease	- MRTX849 - Pembrolizumab - Cetuximab - Afatinib	NCT03785249

AC: AdenoCarcinoma; CRC: ColoRectal Cancer; NSCLC: Non-Small Cell Lung Cancer.

**Table 8 cancers-14-04103-t008:** KRAS inhibition and downstream pathways inhibition(PI3K/CD4/CD6), ongoing studies.

Title	Status	Condition	Intervention	NCT Number
A Study of Abemaciclib (LY2835219) in Participants with Previously Treated KRAS Mutated Lung Cancer (JUNIPER)	Active, not recruiting recruiting	- NSCLC - AC - Stage IV	- Abemaciclib - Erlotinib	NCT02152631
PALBOCICLIB + PD-0325901 for NSCLC & Solid Tumors	Active, not recruiting	- KRAS-mutant NSCLC - Solid Tumors	- Palbociclib - PD-0325901	NCT02022982
A Study to Evaluate the Safety, Pharmacokinetics, and Activity of GDC-6036 Alone or in Combination in Participants With Advanced or Metastatic Solid Tumors With a KRAS G12C Mutation	Recruiting	- NSCLC - CRC - Advanced Solid Tumors	- GDC-6036 - Atezolizumab - Cetuximab - Bevacizumab - Erlotinib - GDC-1971 - Inavolisib	NCT04449874

AC: AdenoCarcinoma; CRC: ColoRectal Cancer; NSCLC: Non-Small Cell Lung Cancer.

**Table 9 cancers-14-04103-t009:** Immunotherapy and KRAS inhibition, ongoing studies.

Title	Status	Condition	Intervention	NCT Number
Rigosertib Plus Nivolumab for KRAS+ NSCLC Patients WhoProgressed on First-Line Treatment	Recruiting	- NSCLC - AC - Stage IV	- Rigosertib - Nivolumab	NCT04263090
A Phase 1/2, Study Evaluating the Safety, Tolerability, PK, and efficacy of Sotorasib in subjects with solid tumors with a specific KRAS mutation (codeBreak 100)	Recruiting	- *KRAS* p.G12C mut - Advanced Solid tumor	- Sotorasib - Anti PD-1/L1 - Midazolam	NCT03600883
Trametinib and Pembrolizumab in Treating Patients With recurrent NSCLC that is metastatic, unresectable or locally advanced.	Active, not recruiting	- Metastatic NSCLC - Stage IIIA-C lung cancer AJCC v8 - Stage IVA-B lung cancer AJCC v8 - Unresectable NSCLC	- Pembrolizumab - Trametinib - Pharmacokinetic study	NCT03225664
Phase 2 Trial of MRTX849 Monotherapy and in Combination With Pembrolizumab for NSCLC With KRAS G12C Mutation KRYSTAL-7.	Recruiting	- Advanced NSCLC - Metastatic NSCLC	- MRTX849 Monotherapy - MRTX849 in Combination with Pembrolizumab	NCT04613596
Pembrolizumab and Trametinib in Treating Patients With stage IV NSCLC and KRAS gene mutaitons.	Active, not recruiting	- *KRAS* Gene Mutation - Metastatic or recurrent NSCLC - Stage IV NSCLC AJCC v7	- Pembrolizumab - Trametinib	NCT03299088
PDR001 in Patients With Non-small Cell Lung Cancer harboring KRAS/NRAS mutation or no actionable genetic abnormalities.	Recruiting	- Cancer - Metastatic Lung Cancer.	- PDR001	NCT03693326
A Study to Evaluate the Safety, Pharmacokinetics, and Activity of GDC-6036 Alone or in Combination in Participants With Advanced or Metastatic Solid Tumors With a KRAS G12C Mutation	Recruiting	- NSCLC - CRC - Advanced Solid Tumors	- GDC-6036 - Atezolizumab - Cetuximab - Bevacizumab - Erlotinib - GDC-1971 - Inavolisib	NCT04449874
Phase 1/2 Study of MRTX849 in Patients With Cancer Having a KRAS G12C Mutation KRYSTAL-1	Recruiting	- Advanced Cancer - Metastatic Cancer - Malignant Neoplastic Disease	- MRTX849 - Pembrolizumab - Cetuximab - Afatinib	NCT03785249
Phase 1/2 Study of MRTX849 in Patients With Cancer Having a KRAS G12C Mutation KRYSTAL-1	Recruiting	- Advanced Cancer - Metastatic Cancer - Malignant Neoplastic Disease	- MRTX849 - Pembrolizumab - Cetuximab - Afatinib	NCT03785249
A Phase 1 Study of the SHP2 Inhibitor BBP-398 in Combination With Nivolumab in Patients With Advanced Non-Small Cell Lung Cancer With a KRAS Mutation	Recruiting	- NSCLC - Solid Tumor	- BBP-398 with Nivolumab	NCT05375084
Study of JDQ443 in patients with advanced solid tumors harboring the KRAS G12 C mutation	Recruiting	- *KRAS* G12C Mutant Solid Tumors - NSCLC - CRC	- JDQ443 - TNO155 - Tislelizumab	NCT04699188
A Study of MK-1084 as Monotherapy and in Combination With Pembrolizumab (MK-3475) in Participants With KRASG12C Mutant Advanced Solid Tumors (MK-1084-001)	Recruting	- Advanced Solid Tumors	- MK-1084 - Pembrolizumab	NCT05067283

AC: AdenoCarcinoma; CRC: ColoRectal Cancer; NSCLC: Non-Small Cell Lung Cancer.

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
