# Peer review of "Molecular Biology and Therapeutic Perspectives for K-Ras Mutant Non-Small Cell Lung Cancers"

_cancers, 2022, doi:10.3390/cancers14174103_

Round 1

Reviewer 1 Report

This is a useful review and generally well written and covers recent developments in KRAS mutant lung cancer. A few recommendations to improve the manuscript:

1. Provide a short description of STK11/LKB1, KEAP1 and NFE2L2/NRF2 and SMARC4.

2. Provide higher resolution Tables.

3. Recommendations for language changes:

28        “fighting a group of NSCLC orphaned by targeted drugs.” Meaning is unclear here. Please rewrite this sentence. Perhaps something like this:  “However, with the approval of a KRAS inhibitor, Sotorasib, the outlook has improved for the previously untargetable KRAS-mutant lung AC which has historically been associated with poorer survival.”

39        “AC patients are generally associated” change to “generally have”

92        “ it is higher for what concerns”   change to something like “ The occurrence of KRAS mutations differs based on the specific lung cancer type/subtype being higher in NSCLC than in SCLC; and within NSCLC KRAS mutations are more frequent in Lung AC than SCC (Table 2).

100      “diffused” change to most “common” or most “prevalent”

123      “diffused” change to most “common” or most “prevalent”

175      “Women empowerment” perhaps change to “women’s liberation movement” (I recommend to keep it in quotation marks in your text”

185      Delete “more in details”

198      Change “familiar history” to “familial history”

235      Change “many proofs suggest” to “much evidence suggests”

256-257 suggestion:   “Though they rarely occur in lung cancer, KRAS mutations combined with EGFR mutations or ALK rearrangement predict poorer response to tyrosine kinase inhibitors”.

298      Change “Authors” to “researchers”

310      Suggestion: The prognosis associated with KRAS mutations depends on co-alterations. A poorer prognosis is found  KRAS mutations co-occur with……

729      “ac-tively” to “actively”

828 Capitalize “In the same study

Author Response

We would like to thank Reviewer 1 for the comments and for the positive evaluation. In the point-by-point letter reported below you can kindly find our answers to the required changes (in red). .

A few recommendations to improve the manuscript:

  1. Provide a short description of STK11/LKB1, KEAP1 and NFE2L2/NRF2 and SMARC4.

According to reviewer’s comment the description of these markers has been implemented (page7, lines 207-209; page8, lines 274-278; page28, lines 862-867).

  1. Provide higher resolution Tables.

We are sorry for the low quality, a new version of the Tables (in WORD format) has been reported in the text.

  1. Recommendations for language changes:

The required language changes have been applied. We thank the reviewer for the precise revision of the text and for suggestions.

28 “fighting a group of NSCLC orphaned by targeted drugs.” Meaning is unclear here. Please rewrite this sentence. Perhaps something like this:  “However, with the approval of a KRAS inhibitor, Sotorasib, the outlook has improved for the previously untargetable KRAS-mutant lung AC which has historically been associated with poorer survival.”

Change on page1, lines 27-29.

39        “AC patients are generally associated” change to “generally have”

Change on page1, lines 38-39.

92        “ it is higher for what concerns”   change to something like “ The occurrence of KRAS mutations differs based on the specific lung cancer type/subtype being higher in NSCLC than in SCLC; and within NSCLC KRAS mutations are more frequent in Lung AC than SCC (Table 2).

Change on page 3, lines 97-99.

100      “diffused” change to most “common” or most “prevalent”

Change on page3, line 111.

123      “diffused” change to most “common” or most “prevalent”

Change on page4, line 121.

175      “Women empowerment” perhaps change to “women’s liberation movement” (I recommend to keep it in quotation marks in your text”

Change on page7, line 195.

185      Delete “more in details”

Change on page7, line 205.

198      Change “familiar history” to “familial history”

Change on page7, line 220.

235      Change “many proofs suggest” to “much evidence suggests”

Change on page8, line 257.

256-257 suggestion:   “Though they rarely occur in lung cancer, KRAS mutations combined with EGFR mutations or ALK rearrangement predict poorer response to tyrosine kinase inhibitors”.

Change on page8, lines 281-283.

298      Change “Authors” to “researchers”

Change on page9, line 323.

310      Suggestion: The prognosis associated with KRAS mutations depends on co-alterations. A poorer prognosis is found  KRAS mutations co-occur with……

Change on page9, line 335-337.

729      “ac-tively” to “actively”

Change on page23, line 742.

828    Capitalize “In the same study

Change on page28, line 854.

Reviewer 2 Report

The Froesch group has written a comprehensive and relevant review article discussing the clinical relevance of KRAS mutation in non-small cell lung cancers (NSCLC). The authors discuss rate of KRAS mutations, other concomitant mutations, KRAS in prediction and prognosis, treatment, resistance and finally interaction between KRAS mutations with immune checkpoint inhibitors. They identified challenges in the field, key among them is using KRAS mutations as a predictive marker for response and identifying cancer in which KRAS is a dependency from those in which it is not essential for survival. They discuss in detail KRAS inhibitors and mechanism of resistance to KRAS inhibitors. Finally, the authors consider potential of improving outcomes by using immune checkpoint inhibitors in KRAS mutant NSCLC. The unique feature about the article is that no such articles has been published before that comprehensively cover roles of KRAS mutations in NSCLC from a clinical standpoint. The review covers all the major recent findings in the field.

One minor comment that needs author's attention are the line 254-256 ‘Recent studies established that in cancers the presence of mutations in different genes simultaneously and the presence of concomitant mutations can influence targeted therapies response or patients’ prognosis [87]’. It is confusing and needs further refinement and additional clarification.

Author Response

We would like to thank Reviewer 2 for the overall positive evaluation of our study. Below, you can kindly find our answer to the required change in red.

One minor comment that needs author's attention are the line 254-256 ‘Recent studies established that in cancers the presence of mutations in different genes simultaneously and the presence of concomitant mutations can influence targeted therapies response or patients’ prognosis [87]’. It is confusing and needs further refinement and additional clarification.

This sentence has been modified and we hope is now more comprehensible (page8, lines 279-281).
